# The breath shape controls intonation of mouse vocalizations

**Alastair MacDonald[1], Alina Hebling[2], Xin Paul Wei[1,3], Kevin Yackle[1]***

[1]Department of Physiology, University of California-San Francisco, San Francisco, United States; [2]Neuroscience Graduate Program, University of California-San Francisco, San Francisco, United States; [3]Biomedical Sciences Graduate Program, University of California-San Francisco, San Francisco, United States

**Abstract** Intonation in speech is the control of vocal pitch to layer expressive meaning to communication, like increasing pitch to indicate a question. Also, stereotyped patterns of pitch are used to create distinct sounds with different denotations, like in tonal languages and, perhaps, the 10 sounds in the murine lexicon. A basic tone is created by exhalation through a constricted laryngeal voice box, and it is thought that more complex utterances are produced solely by dynamic changes in laryngeal tension. But perhaps, the shifting pitch also results from altering the swiftness of exhalation. Consistent with the latter model, we describe that intonation in most vocalization types follows deviations in exhalation that appear to be generated by the re-activation of the cardinal breathing muscle for inspiration. We also show that the brainstem vocalization central pattern generator, the iRO, can create this breath pattern. Consequently, ectopic activation of the iRO not only induces phonation, but also the pitch patterns that compose most of the vocalizations in the murine lexicon. These results reveal a novel brainstem mechanism for intonation.

## eLife assessment

This **important** study examines the relationship between expiratory airflow and vocal pitch in adult mice during the production of ultrasonic vocalizations and also identifies a molecularly defined population of brainstem neurons that regulates mouse vocal production across development. The evidence supporting the study's conclusions that expiratory airflow shapes vocal pitch and that these brainstem neurons preferentially regulate expiratory airflow is novel and **compelling**. This work will be of interest to neuroscientists working on mechanisms and brainstem circuits that regulate vocal production and vocal-respiratory coordination.

## Introduction

Modulation of the frequency of produced sound, perceived as pitch, creates meaning within words or phrases through intonation (*Prieto, 2015*). For example, in English, an increasing pitch is used to indicate a question or stress importance and a decreasing pitch communicates a declaration. Additionally, the concatenation of specialized sounds accented by variations in pitch enriches the composition of spoken language, like how the same sound at divergent pitches can relay different meanings (*Howie, 1976*). Two key pieces of the phonation system are the larynx (the 'voice box') and the breathing muscles (*Berke and Long, 2009*; *Laplagne, 2018*). Succinctly, the breathing muscles drive airflow through a narrowed larynx to produce a basic vocalization (*Finck and Lejeune, 2009*). The local speed of the airflow through the larynx dictates the fundamental frequency of the tone, so changes in either the swiftness of the breath exhalation or the extent of laryngeal closure can both, presumably, alter the pitch (*Kelm-Nelson et al., 2018*, *Herbst, 2016*, *Mahrt et al., 2016*). While control of the size of

*For correspondence:
Kevin.Yackle@ucsf.edu

**Competing interest:** The authors declare that no competing interests exist.

the laryngeal opening is well established as a mechanism to regulate the dynamic changes in pitch for human to rat and mouse vocalizations (*Titze et al., 1989*; *Johnson et al., 2010*; *Riede et al., 2017*), the contribution of exhalation itself remains to be carefully defined. In fact, it is presumed that the velocity of expiration only modulates the vocal amplitude or loudness (*Riede, 2011*; *Riede, 2013*). This perception stems from the airflow of the rodent breath not strongly predicting the pitch. Yet paradoxically, an injection of air below the larynx to enhance flow increases pitch (*Riede, 2011*). This incongruity even extends to songbirds, a leading vocalization model system (*Suthers et al., 2002*; *Schmidt and Martin Wild, 2014*; *Plummer and Goller, 2008*; *Goller and Cooper, 2004*). Here, we seek to resolve this inconsistency by taking advantage of the experimental, behavioral, and genetic approaches in the mouse (*Yackle, 2023*). If two independent variables are used to alter pitch, like the larynx and the breath airflow, then the interplay would enhance the ability to produce a diverse repertoire of sounds and thereby enable a broader lexicon.

The medullary brainstem possesses at least two means that might account for the two control points proposed above, laryngeal diameter and exhalation speed. First, modulation of some laryngeal premotor neurons in the retroambiguus (RAm) and perhaps intermingled or nearby motor neurons modulates the size of the laryngeal opening (*Kelm-Nelson et al., 2018*; *Hage, 2009a*; *Park et al., 2024*) and is even sufficient to evoke vocalizations across species, albeit mostly abnormal (*Zhang et al., 1995*; *Hartmann and Brecht, 2020*; *Veerakumar et al., 2023*; *Park et al., 2024*). Second, the vocalization central pattern generator (CPG) we recently described, called the intermediate reticular oscillator (iRO), induces coordinated changes in the expiratory airflow and laryngeal closure (*Wei et al., 2022*). For example, during neonatal cries, the iRO oscillates exhalation speed and larynx activity to time the syllable sounds. Thus, the RAm provides a mechanism to modulate pitch by controlling laryngeal diameter independently from the iRO altering the tone by dictating the speed of the breath expiratory airflow. While the contribution of RAm in adult phonation has recently been established in mice (*Veerakumar et al., 2023*; *Park et al., 2024*), the role of the iRO remains undefined.

Here, we describe the coordinated changes in breath airflow and pitch in the 10 vocalizations of the adult murine lexicon (*Grimsley et al., 2011*). We describe that the modulation of pitch for the different vocalizations either correlates or anti-correlates with the changes in exhalation velocity. These results support a model in which two independent mechanisms involving changes in laryngeal opening airflow control tone. Consistently, we found that the pattern of breathing muscle activity is different for each mechanism. For example, the muscle that drives the inspiration phase of the breath is ectopically re-engaged to break expiratory airflow and decrease pitch. This mirrors how the iRO rhythmically patterns the breathing muscles to time syllables during neonatal cries. Using anatomical, molecular, and functional approaches, we demonstrate that the iRO vocal CPG drives changes in breath expiratory airflow to pattern pitch and produce seven of the ten vocalizations in the endogenous lexicon. These data resolve the prior paradoxical role of exhalation speed in sound production and show it can directly control pitch. Additionally, we establish the iRO as a mechanistic basis for intonation. And lastly, these results generalize the crucial role of the iRO in phonation across developmental stages and, we presume, across species.

## Results

### Vocalizations are produced by a program coupled to breathing

It is possible that the 10 murine ultrasonic vocalizations (USVs) defined by unique pitch patterns ('syllable types'; *Grimsley et al., 2011*) are formed by distinct breaths or as substructures nested in a common breath. Prior work has suggested the latter (*Sirotin et al., 2014*). To expand upon this, we simultaneously measured breathing and USVs by customizing the lid of a whole-body plethysmography chamber to accommodate a microphone. Male mice in the chamber were exposed to fresh female urine and robustly sniffed and vocalized for the first 5–10 min of the recording at a peak rate of about 4 events per second (*n*=6) (*Figure 1A and B*). A vocalization was classified as a narrow-band sound in the 40–120 kHz ultrasonic frequency range during a single breath (*Figure 1A*). The instantaneous frequency of vocalization breaths was typically between 5 and 10 Hz (mean: 7.5 Hz) (*Figure 1C*) and mostly occurred during episodes of rapid sniffs (8.5–10 Hz) (*Figure 1A and B*), as previously reported (*Sirotin et al., 2014*; *Castellucci et al., 2018*). When compared to neighboring breaths, vocalization breaths were slightly slower overall (*Figure 1C*) with subtly larger inspiratory and perhaps

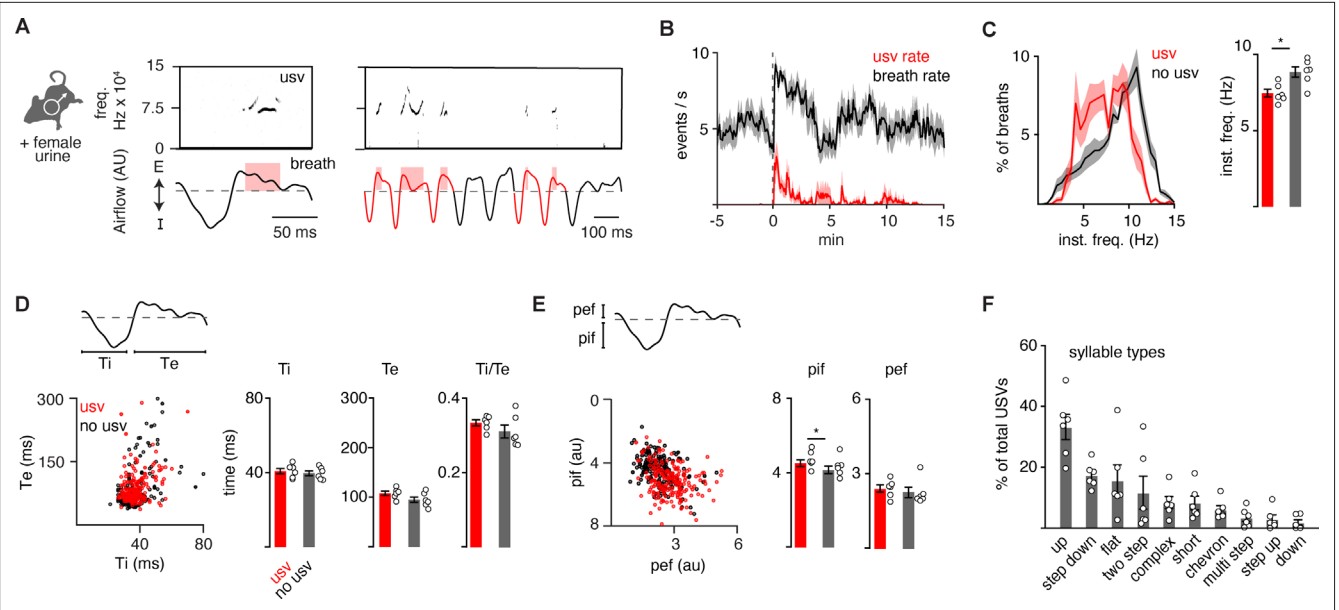

**Figure 1.** The full repertoire of vocalizations occurs within a normal appearing breath. (**A**) Male mice exposed to female urine produced ultrasonic vocalizations (USVs) at about 75 kHz (top) that coincide with the expiratory airflow (E, arbitrary units) of the breath cycle (bottom). Red box indicates the length of the USV. A bout of vocalizations contains breaths with USVs (red) interspersed among sniff breaths (black). (**B**) Rates of breathing (black) and USV production (red). Exposure to female urine at time 0, *n*=6 mice. (**C**) Left, histogram of the instantaneous frequency of breaths with and without USVs from *n*=6 animals. Right, average instantaneous frequency for each mouse (mean ± SEM). Each dot is the mean from each animal. p-Value 0.03; paired t-test. (**D**) Scatter plot of the inspiratory (Ti) and expiratory time (Te) for USV (red) and non-USV (black) breaths from a representative animal. Right, bar graph of mean ± SEM of Ti, Te, and the ratio for *n*=6. p-Values 0.40, 0.18, and 0.25; paired t-test. (**E**) The breath peak inspiratory (pif) and expiratory (pef) airflow represented as in **D**. p-Values 0.01, 0.27; paired t-test. (**F**) Bar graph (mean ± SEM) of the percent of total USVs for each type from *n*=6 mice.

The online version of this article includes the following source data and figure supplement(s) for figure 1:

**Source data 1.** Characterization of basal versus ultrasonic vocalization (USV) containing breaths.

**Figure supplement 1.** Ultrasonic vocalization (USV) onset and offset during expiration.

**Figure supplement 2.** Representative example of the most common ultrasonic vocalization (USV) types and the onset and offset times during expiration.

**Figure supplement 3.** Representative example of the many ultrasonic vocalization (USV) types and the onset and offset times during expiration.

**Figure supplement 4.** Raster plot of ultrasonic vocalization (USV) on- and offset plotted upon the breathing rhythm.

expiratory airflow despite similar durations of each phase (*Figure 1D and E*). These data reveal that a vocalization breath appears mostly like a normal breath but with the addition of a nested sound pattern. This led us to hypothesize that a distinct sub-program is activated within a breath to generate a vocalization.

The adult murine lexicon is composed of at least 10 USV syllable types that are defined by different, but stereotyped, patterns of pitch. Most breaths contain a single syllable (88%), which we define as a continuous USV event (*Figure 1F* and *Figure 1—figure supplements 2–3*), however, on some occasions, we observed two (8%) or three (4%) syllables separated by >20 ms within a single breath (*Figure 1—figure supplement 3*). This structure mirrors that described in neonatal cries (*Wei et al., 2022*). A pre-trained convolutional neural network (CNN) was used to classify USVs into different types based on changes in pitch (*Fonseca et al., 2021*), and the on- and offset of each vocalization was overlayed upon the corresponding breath airflow (*Figure 1—figure supplements 1–3*). Vocalizations began and ended throughout expiration (*Figure 1—figure supplement 1*), and the most common tended to start near the onset of exhalation and ended shortly thereafter (like the up frequency modulated [fm], step down, flat, and short types) (*Figure 1—figure supplement 2*). Vocalizations with more intricate changes in pitch had more variable times of on- and offset (like complex, chevron, two step, multi, step up, down fm) (*Figure 1—figure supplement 3*). And lastly, when the vocalizations occurred late in expiration, the duration of this breath phase was prolonged (*Figure 1—figure*

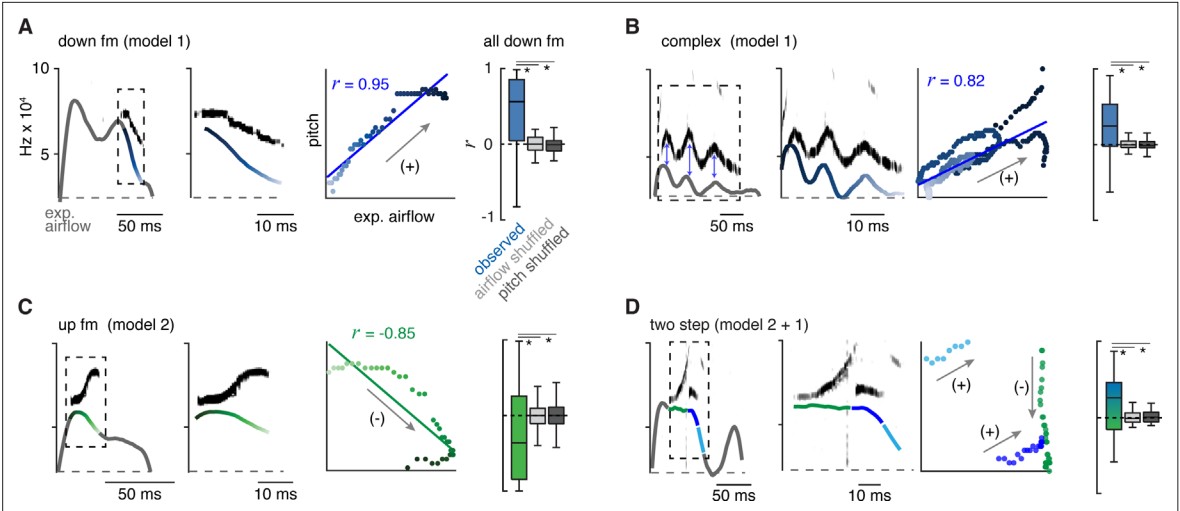

**Figure 2.** The 10 types of ultrasonic vocalizations (USVs) are produced by at least two mechanisms that modulate airflow. (**A**) Left, example of the expiratory airflow and pitch for a down frequency modulated (fm) USV. Middle, magnification of airflow and sound. The scale of airflow is not displayed. The time of breath airflow from expiration onset during the USV is color coded blue to white. Scatter plot of instantaneous expiratory airflow and pitch for the single USV and the correlation (line, *r*). Note, the change in pitch mirrors airflow (annotated as '+'), consistent with the 'breath modulation' called model 1. Box and whisker plot of *n*=40 down fm correlation coefficients (*r*). Controls, normal USV pitch vs shuffled airflow or shuffled pitch to normal USV expiratory airflow. (**B–D**) Representative expiratory airflow and pitch, box and whisker plot of all *r* values, and onset/offset for complex (*n*=165), up fm (*n*=589), and two step (*n*=61) vocalizations represented as in **A**. The airflow for each unique USV element is uniquely color coded as green, blue, or purple. Note, the change in pitch for two components correlates and one anti-correlates. This is consistent with both mechanisms being sequentially used. Annotated as mixed blue and green box and whisker plot. *=p<0.05, one-way ANOVA with Sidak's post hoc test.

The online version of this article includes the following source data and figure supplement(s) for figure 2:

**Source data 1.** Observed and shuffled correlations between pitch and expiratory airflow.

**Figure supplement 1.** Correlation coefficient and onset/offset time for six ultrasonic vocalizations (USVs).

*supplement 4*). The bias of USV timing by the breath combined with the USV modulation of breath length demonstrate these programs are independent but reciprocally coupled.

## Two mechanisms create the changes in pitch pattern

Fluctuations in airflow speed through the larynx produce changes in the sound's pitch. For example, augmenting airflow through the explanted rodent larynx increases pitch (*Mahrt et al., 2016*). We proposed two potential mechanisms that could contribute to how the laryngeal airflow is modulated to form the distinct USV types in the murine lexicon: one based on the swiftness of exhalation pushing air through the larynx (model 1), and the other based on the diameter of the laryngeal opening (model 2). According to the first model, changes in pitch positively correlate with the breath expiratory airflow measured by plethysmography, which we term positive intonation. On the other hand (model 2), a narrowed larynx increases pitch by speeding local airflow while simultaneously impeding the overall expiratory airflow measured by plethysmography; we call this negative intonation. Note, these models can form similar expiratory airflow patterns, but predict opposite relationships to pitch.

We assessed for evidence of each model by calculating the correlation coefficient (*r*) between instantaneous expiratory airflow and the corresponding USV fundamental frequency. Down or up fm USVs served as simple USV examples and we found that these were positively or negatively correlated, respectively (median *r*=0.62 and –0.46, *Figure 2A and C*). These two were also distinguished by the sound onset and offset, whereby down fm started and ended later during the expiration (*Figure 1—figure supplements 2–3*). These distinguishing features are consistent with the sounds being produced by separate mechanisms to alter pitch, positive and negative intonation.

Between these mechanisms, most syllable types displayed positive intonation. Five of the other eight USV syllable types had positively shifted intonation, the complex, step down, chevron, two step, and multi (median *r*=0.31, 0.28, 0.32, 0.19, and 0.24, respectively) (*Figure 2B and D* and *Figure 2— figure supplement 1*). In particular, the mirrored oscillations in the breath expiratory airflow and pitch

during a complex vocalization best illustrated the positively coupled relationship (model 1, *Figure 2C*). In contrast, the up fm was the only USV with negative intonation. The *r* values for positively and negatively correlated USVs were not random, as they fell outside *r* values computed in simulated datasets composed of USVs with shuffled expiration airflow or expirations with shuffled pitch frequencies (*Figure 2* and *Figure 2—figure supplement 1*).

The two step and step up USVs (median *r*=0.19 and –0.03) appeared to have a portion of the pitch pattern correlated with the expiratory airflow, while the other part(s) were un- or anti-correlated (e.g. the two step, *Figure 2E*). This suggests that the pitch is produced by switching between positive and negative intonation mechanisms within the breath. The remaining two USV types (flat and short) occurred with various breath shapes which resulted in a wide range of *r* values (*Figure 2—figure supplement 1*). Some USV syllable types are defined by the presence of large, instantaneous jumps in frequency (like the step up, step down, two- and multi-step). A jump was not associated with a corresponding change in airflow. Also, the jumps did not predict the subsequent intonation pattern. For example, step up syllables could be negative to negative (20%), negative to positive (20%), or positive to positive (60%) intonations.

Across all USV types, we found that the relationship between airflow and pitch is relative rather than absolute which is expected since the relationship is determined by at least two independent variables (laryngeal tone and exhalation speed). In summary, these results establish an important positive connection between the breath expiratory airflow to modulate pitch (model 1). This supports the hypothesis that a vocalization pattern generator must integrate with and even control the breath airflow as a key mechanism to produce various USV types in the murine lexicon.

## Inspiratory and laryngeal muscles have coordinated activity that represents positive and negative intonation

To explore the mechanisms underlying the two intonation models, we simultaneously recorded the electrical activity of the primary muscles for breathing, the diaphragm, and laryngeal control (thyroarytenoid and cricothyroid) during basal breathing and bouts of vocalizations in male mice. Electromyography (EMG) electrodes were permanently placed along the diaphragm and inserted into the larynx, while breathing and USVs were measured in parallel by whole-body plethysmography and a microphone (as in *Figure 1*). Electrocardiogram signals were annotated and removed from the diaphragm EMG recordings post hoc. According to our models and the timing of the syllable types within the breath (*Figure 1—figure supplements 2–3*), we anticipated that USVs with positive intonation would have a coordinated re-activation of the diaphragm and laryngeal muscles later in expiration, while the up fm would only have laryngeal activity at expiration onset.

EMG activity during basal breathing displayed the three phases of the breath cycle (*Yackle, 2023*). The diaphragm was active during inspiration and larynx in the subsequent post-inspiration period, and these were followed by a late expiration phase where neither muscle was active (*Figure 3A*). The airflow measured by plethysmography had an ~10 ms lag when compared to the EMG activities (blue and red arrows, *Figure 3A*). Upon exposure to female odors, the male mice produced bouts of USVs containing the entire lexicon for several minutes, demonstrating that electrodes interfered with neither breathing nor sound production (*Figure 3B*). Breaths containing USVs were distinguished from adjacent breaths lacking sound by an increase in laryngeal muscle activity that prolonged into expiration. Like airflow, the sound followed EMG activity by ~10 ms. More than 10 mice were studied and, unsurprisingly, given the invasiveness of the EMGs only three produced robust signals and the number of acquired vocalizations varied from tens to thousands between these mice (*n*=70, 1482, 2819). The analysis below is from a single animal that had the clearest EMG signals, but the same results and relationships were observed in all three animals.

To study positive intonation, we analyzed the down fm and complex USVs. Down fms started later during expiration and began at a drop in expiratory airflow (*Figure 3C*). The diaphragm activity occurred just before the decrease in expiratory airflow (blue arrow, *Figure 3C*) and the following laryngeal activity persisted during the sound. This pattern reflects the activity of the muscles during a normal breath, inspiration (diaphragm) then post-inspiration (larynx), but this pattern was ectopically embedded within an expiration. Similarly, complex vocalizations had the diaphragm then laryngeal activity pattern during expiration, but this cycled multiple times concurrent with the increase and decrease in expiratory airflow and pitch (*Figure 3C*). Complex USVs had an average of 1.5 cycles per

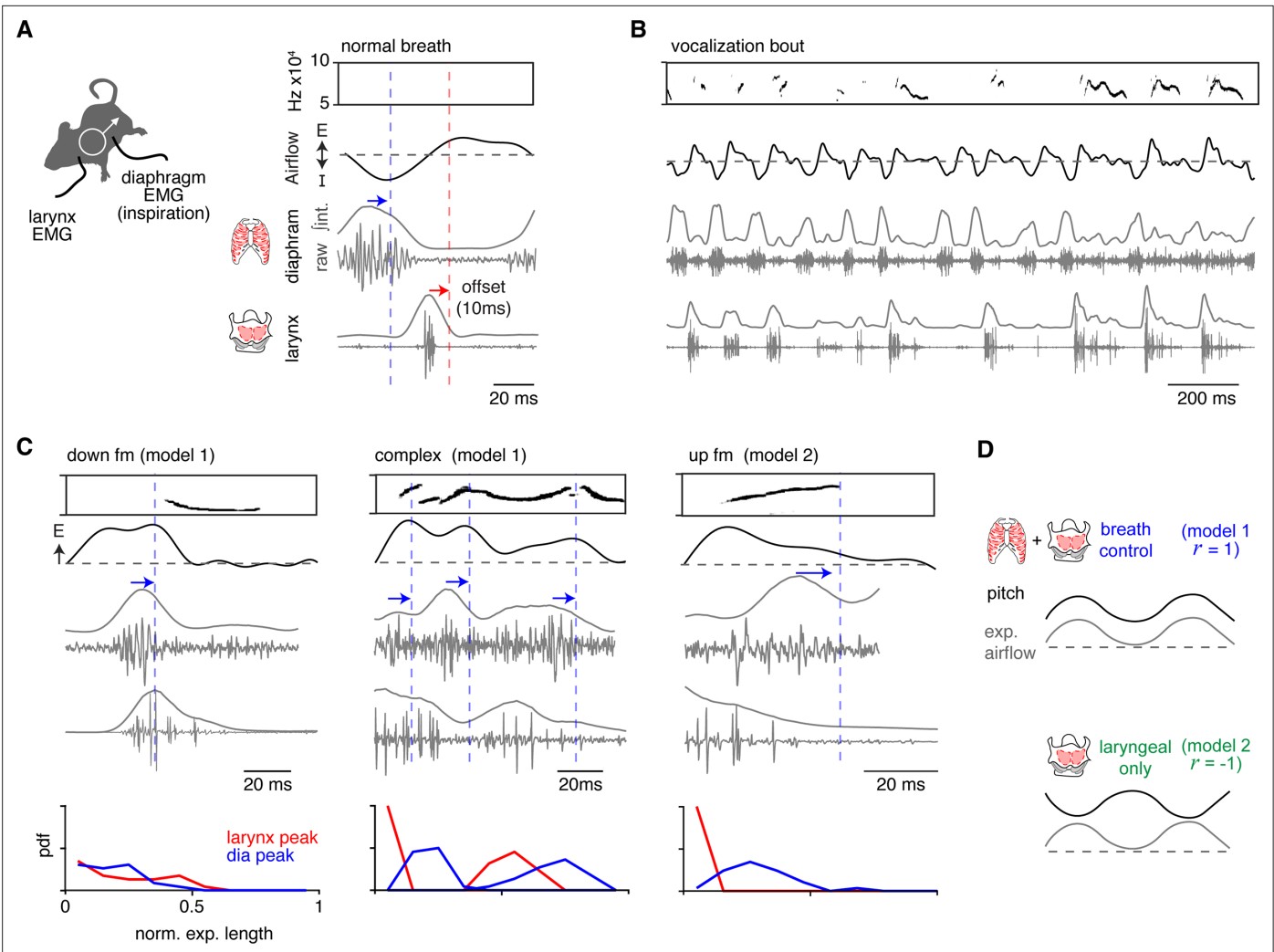

**Figure 3.** Ectopic activation of inspiratory and laryngeal muscles corresponds to changes in vocalization pitch. (A) Activity of the diaphragm (inspiratory) and laryngeal (thyroarytenoid and cricothyroid) muscles were recorded in vivo by electromyography (EMG) simultaneously with breathing and ultrasonic vocalizations (USVs). Right, example of sound, breath airflow, and muscle activities during a basal breath. The diaphragm shows restricted activity during inspiration and laryngeal muscles are active during the post-inspiration period. Note, blue and red arrows/lines indicate the ~10 ms offset of the peak EMG activity and airflow/sound measurements. (B) Representative vocalization bout shows robust vocalizations and breathing in mice with implanted EMGs. (C) Representative sound, airflow, diaphragm, and laryngeal EMGs during the expiration of a down frequency modulated (fm) (n=23 annotated), complex (n=43 annotated), and up fm USV (n=29 annotated). Blue arrow and dashed line indicate the airflow and sound offset from the diaphragm peak muscle activity. Bottom, probability density function (PDF) for the peak of the integrated EMG activity for the diaphragm (blue) and larynx (red) during the normalized expiration. *Y*-axis is from 0 to 1. Note, data in A–D is from one animal with clear EMG signals that represents the findings in all three animals studied. (D) Schematics for each model. Model 1 – breath control: inspiratory and laryngeal muscles have alternating activity throughout the sound/expiration and a *r*>0 for pitch vs expiratory airflow. Muscle activities correspond to an increase (laryngeal) and a decrease (diaphragm) in pitch. Model 2 – laryngeal only: laryngeal but not diaphragm activity occurs during the sound and produces a *r*<0 for pitch vs expiratory airflow.

The online version of this article includes the following source data for figure 3:

**Source data 1.** Diaphragm and laryngeal electromyography (EMG) peaks normalized to expiratory length.

expiration, the interval between diaphragm bursts was 43±14 ms, and the laryngeal activity occurred at 69 ± 12% through this diaphragm-to-diaphragm interval. Also, in ~19% of the complex USVs, sound, laryngeal, and diaphragm activity co-occurred, suggesting that other mechanisms contribute to the diversity of sounds.

Up fm represents negative intonation, and correspondingly, the activity of the muscles distinct from the positively intonated types. At the peak of the post-inspiratory period of the breath, the

vocalization began and persisted throughout the laryngeal muscle activity (*Figure 3C*). The sound ended at the onset of a burst in the diaphragm EMG (*Figure 3C*).

In summary, these EMG studies of the key inspiratory muscle and larynx serve to reflect the core components of the breathing CPG that produce inspiration and post-inspiration. Ectopic activation of these antiphase patterns during the expiration of a vocal breath appears to result in a USV with positive intonation and the cyclic engagement of this leads to an oscillating pitch. Additionally, the termination of the USV with negative intonation corresponds to the re-activation of inspiratory muscles (*Figure 3D*). The novel finding that the endogenous pattern of the breathing CPG is re-engaged within an adult vocal breath, a 'mini-breath', mimics the rhythmic syllables of the neonatal vocalizations patterned by the iRO (*Wei et al., 2022*). This suggests that the iRO also plays a central role in the production of adult vocalizations.

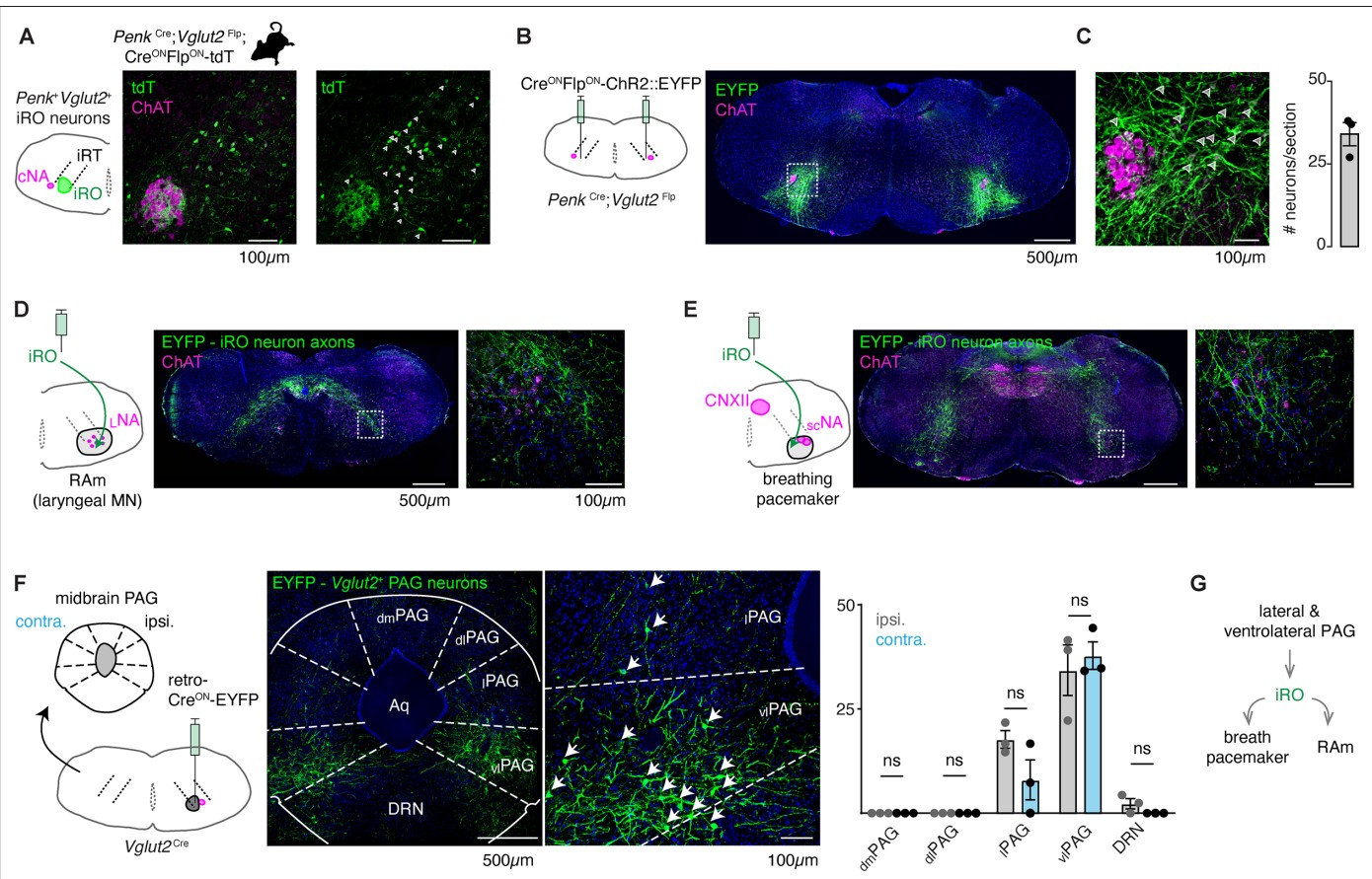

**Figure 4.** Anatomically and molecularly defined intermediate reticular oscillator (iRO) neurons form a brainstem phonatory circuit. (**A**) Labeling of *Penk+Slc17a6+* neurons in the iRO anatomical region in adult *Penk*-Cre;*Slc17a6*-Flp;Ai65 mice (Cre$^{ON}$Flp$^{ON}$-tdTomato) (observed in *n*=5 mice). The iRO region is defined as medial to the compact nucleus ambiguus (cNA, ChAT+) in the ventral intermediate reticular formation (iRT). Note, the cNA is filled with tdTomato labeled axons. Cell bodies marked with arrowhead. (**B**) Bilateral stereotaxic injection of AAV Cre$^{ON}$Flp$^{ON}$-ChR2::EYFP into the iRO anatomical region of *Penk*-Cre;*Slc17a6*-Flp adult mice. (**C**) Magnified boxed region in **B**. Arrowheads label neuron soma quantified right (*n*=3). (**D**) Axons of EYFP expressing iRO neurons from **B** in the retroambiguus (RAm) anatomical region where laryngeal premotor and motor neurons are located. (**E**) Axons of EYFP expressing iRO neurons from **B** in the breathing pacemaker. (**F**) Unilateral retrograde AAV Cre$^{ON}$-EYP (AAVrg) stereotaxic injection into the iRO region in *Slc17a6*-Cre adults (*n*=3). Glutamatergic neurons were identified in the contralateral (contra.) and ipsilateral (ipsi.) midbrain periaqueductal gray (PAG). Anatomical regions of the PAG: dorsomedial (dm), dorsolateral (dl), lateral (l), ventrolateral (vl) nearby to the dorsal raphe nucleus (DRN) and surrounding the cerebral aqueduct (Aq). Quantification of glutamatergic PAG neurons in each region demarcated, ns = not statistically significant; two-way ANOVA with Sidka's post hoc test. (**G**) Model schematic of the iRO as a central component of the brainstem phonation circuit to convert a vocalization 'go' cue from the PAG into a motor pattern.

The online version of this article includes the following source data for figure 4:

**Source data 1.** Quantification of periaqueductal gray (PAG) histology.

## The iRO resides within the adult brainstem phonation circuit

The iRO has yet to be identified in adult mice. The iRO is molecularly defined in the neonate by the co-expression of *Preproenkephalin* (*Penk*) and *Vesicular glutamate transporter 2* (*Slc17a6*) and is anatomically localized to the medullary ventral intermediate reticular formation (iRT) directly medial to the compact nucleus ambiguus (NA) (*Wei et al., 2022*). We determined that the iRO molecular and anatomical features exist in adults in two ways. First, we generated triple transgenic mice that label *Penk*+*Slc17a6*+ neurons and the derived lineages with tdTomato (*Penk*-Cre; *Slc17a6*-Flp; Ai65) (*Figure 4A*). And second, we stereotaxically injected the iRO region of *Penk*-Cre; *Slc17a6*-Flp mice with a Cre- and Flp-dependent reporter adeno-associated virus (AAV Cre$^{ON}$Flp$^{ON}$-ChR2::YFP) (*Figure 4B and C*). Consistent with the definition of the iRO in neonatal mice, tdTomato+ and YFP+ *Penk*+*Slc17a6*+ neurons were found in the iRT adjacent to the compact NA (*Figure 4A–C*). These results demonstrate that the ventrolateral medulla of adult mice contains neurons with the molecular and anatomical identity of the iRO.

Neonatal iRO neurons are presynaptic to the kernel of breathing, the pacemaker for inspiration (preBötzinger complex [preBötC]) (*Smith et al., 1991*), and premotor to multiple laryngeal and tongue muscles. We traced the YFP+ axons of *Penk*+*Slc17a6*+ neurons (*Penk*-Cre; *Slc17a6*-Flp and AAV Cre$^{ON}$Flp$^{ON}$-ChR2::YFP) and found they elaborated within the NA and RAm where laryngeal premotor and motor neurons localize (*Figure 4A and D*), the breathing pacemaker (*Figure 4E*), and the hypoglossal (tongue) motor nucleus (*Figure 4E*). The projection patterns of these *Penk*+*Slc17a6*+ neurons provide additional evidence that these adult neurons maintain the same connectivity properties as the neonatal iRO neurons, indicating they can control the key elements for vocalization: the breath airflow and larynx.

In adult mice, vocalizations have been triggered by activation of the midbrain periaqueductal gray (PAG), namely glutamatergic neurons in the lateral to ventrolateral subregion (*Michael et al., 2020*; *Chen et al., 2021*; *Tschida et al., 2019*). Note, an exact definition of PAG-USV stimulating neurons remains to be satisfactorily described. To assess if the iRO region is positioned downstream of the ventrolateral PAG, we unilaterally injected *Slc17a6*-Cre mice with a Cre$^{ON}$-ChR2::YFP expressing retrograde traveling AAV (AAVrg) (*Figure 4F*). Among the labeled brain regions, we found YFP+ neurons in a region of the midbrain PAG overlapping with areas that contain PAG-USV neurons. To our surprise, neurons from the ipsi- and contralateral PAG projected to the iRO region in nearly equal numbers (*Figure 4F*). These molecular, anatomical, and neural morphology characterizations reveal that the iRO exists in adults and is embedded within the brainstem phonation network (PAG → iRO → the preBötC, NA, RAm, hypoglossal) (*Figure 4G*).

## Ectopic activation of the putative iRO-induced vocalization

If these labeled *Penk*+*Slc17a6*+ neurons are indeed the iRO, we anticipated that ectopic activation would induce vocalization. We tested this in two ways. First, we generated *Penk*-Cre;*Slc17a6*-Flp;Cre$^{ON}$Flp$^{ON}$-ReaChR triple transgenic mice which express the red-shifted Channel Rhodopsin in *Penk*+;*Slc17a6*+ neurons and the derived lineage (ReaChR mice) and second, we stereotaxically injected the AAV Cre$^{ON}$Flp$^{ON}$-Channel Rhodopsin2::YFP (ChR2) into the iRO region of *Penk*-Cre;*Slc17a6*-Flp mice. In both instances we implanted optic fibers above the iRO bilaterally to further localize neural activation (*Figure 5A* and *Figure 5—figure supplement 1A*). In both experimental regimes, ectopic light activation of the *Penk*+*Slc17a6*+ neurons induced bouts of vocalizations where the breathing rate was entrained by the frequency of stimulation (*Figure 5A*, *Figure 5—figure supplement 1I*). Most bouts and the breaths within contained vocalizations (*Figure 5B* and *Figure 5—figure supplement 1B*) and the amplitudes of all elicited breaths were significantly increased (*Figure 5—figure supplement 1J*). Some AAV-ChR2 mice showed previously described broad-band harmonic vocalizations (like *Grimsley et al., 2011*), while others did not vocalize (*n=5/9*), likely due to incomplete labeling. Note, the audible sound during inspirations in these animals reflects orofacial movement artifacts that result in the fiberoptic colliding with the plethysmography chamber walls. Additionally, the ReaChR animals without vocalizations were found to have 'off-target' optic fiber implants (*n=2/6*). Taken together, these data are consistent with the notion that the iRO is sufficient to induce phonation via control of both breath airflow and laryngeal opening, just as it does in neonatal cries.

To demonstrate the specialization of the iRO neurons for vocalization and the inability of modulated breathing alone to elicit USVs, we performed several additional control experiments. First,

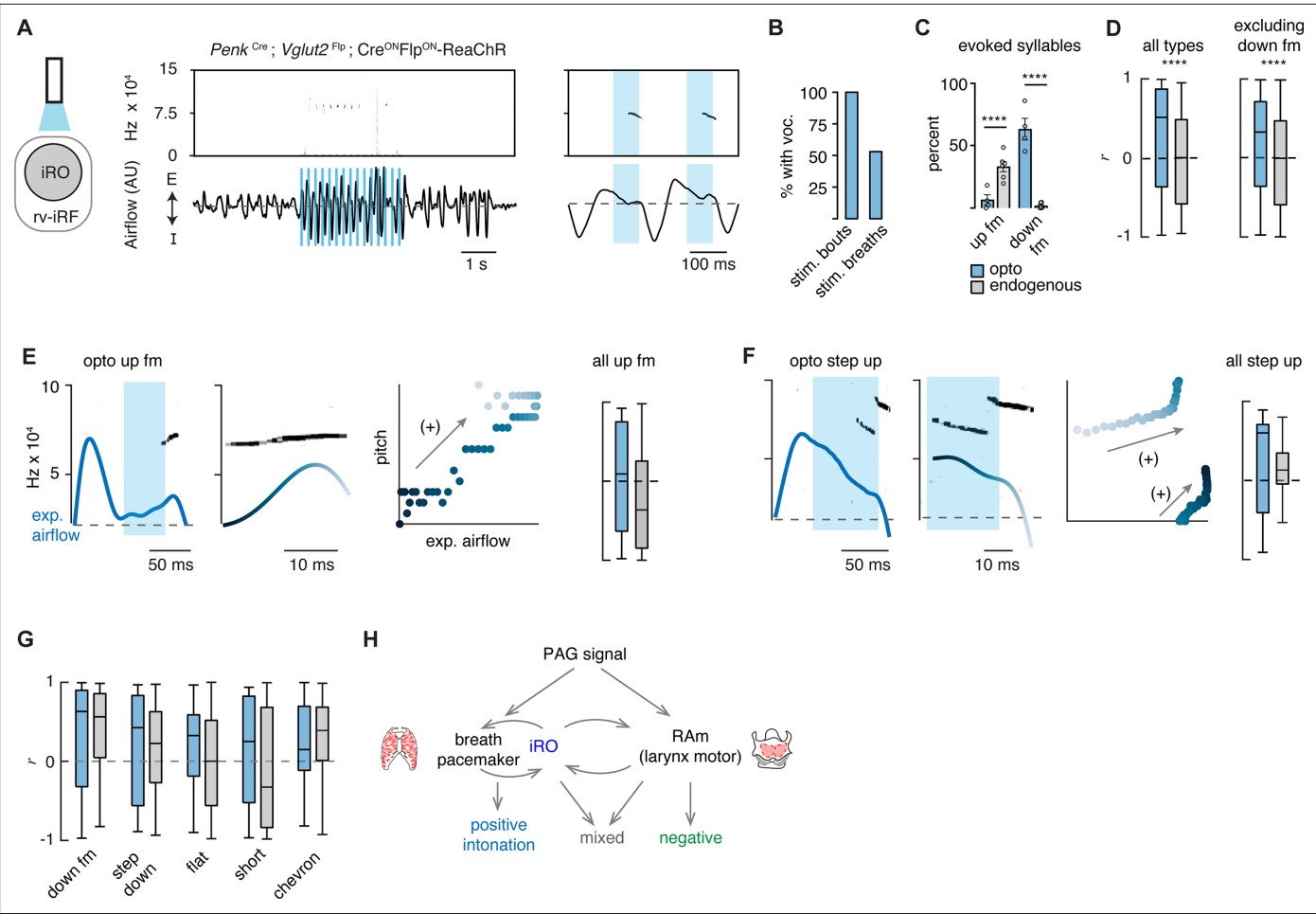

**Figure 5.** Ectopic activation of the intermediate reticular oscillator (iRO) evokes airflow correlated ultrasonic vocalization (USV) types and switches the relationship of the anti-correlated types. (**A**) Optogenetic activation of the iRO region in *Penk*-Cre;*Slc17a6*-Flp;CreᴼᴺFlpᴼᴺ-ReaChR mice evoked USVs (blue box, 5 Hz stimulation). USVs occurred during, or shortly after laser onset. (**B**) Percentage of stimulation bouts containing at least one USV and the percentage of breaths within the stimulation window containing a USV. (**C**) Percentage of optogenetically evoked (blue) or endogenously occurring (gray) syllables that are up frequency modulated (fm) of down fm. ****p-value<0.001; two-way ANOVA with Sidak's post hoc test, p>0.05 for all other types. (**D**) Box and whisker plot of the correlation coefficient between breathing airflow and pitch (*r*) for all opto evoked (*n*=395) and endogenous (*n*=1850) USVs and all opto evoked and endogenous vocalizations without down fm (*n*=143 and 1810) from *n*=4 opto and n=6 endogenous mice. ****p-value<0.001; Mann-Whitney test. (**E**) Left, example of the expiratory airflow and pitch for an optogenetically evoked up fm USV. Middle, magnification of airflow and sound. Time of breath airflow during the USV is color coded blue to white. Scatter plot of instantaneous expiratory airflow and pitch for the single USV. Compare to endogenous up fm USV in *Figure 2C*. Right, box and whisker plot of correlation coefficients (*r*) for each optogenetically evoked and endogenous up fm USV (*n*=15 vs 589). (**F**) Step up USV as in **E**. Box and whisker plot of correlation coefficients (*r*) for each optogenetically evoked and endogenous step up USV (n=15 vs 40). (**G**) Box and whisker plots for the remaining *r*-values of optogenetically evoked USV types. Down fm n=242 vs 40, step down n=38 vs 293, flat n=34 vs 337, short n=27 vs 168, and chevron n=10 vs 99 from n=4 opto and 6 endogenous mice. (**E–G**) Two-way ANOVA with Sidak's post hoc test for two-way comparisons was used; all p-values>0.05. (**H**) Schematic illustrating the two mechanisms to pattern the USV pitch. Left, the reciprocal connection between the iRO and breathing pacemaker patterns the USVs with a positive correlation between pitch and airflow. Right, the retroambiguus (RAm) control of the larynx dictates the anti-correlated USV types. Middle, the combination of these two mechanisms within a single breath create additional USV patterns.

The online version of this article includes the following source data and figure supplement(s) for figure 5:

**Source data 1.** Comparison between ultrasonic vocalization (USV) type and intonation correlations for optogenetically evoked and endogenous USVs.

**Figure supplement 1.** Optogenetic modulation of breathing and ultrasonic vocalizations (USVs) for different molecularly defined cell types in the intermediate reticular oscillator (iRO) anatomical region.

to ensure that just stimulation of breathing is insufficient to elicit vocalization, we optogenetically excited the glutamatergic preBötC neurons (*Slc17a6*-Cre with AAV Cre$^{ON}$-ChR2). Indeed, we found that, although breathing sped up, optogenetic stimulation never elicited vocalizations (*Figure 5—figure supplement 1C, H, I*). And second, to determine if the ability to elicit vocalizations was generalizable to other neural types in the iRO anatomical region, we activated *Penk*$^+$, *µ-opioid receptor*$^+$*Slc17a6*$^+$, *Tachykinin 1*$^+$, and *Vesicular GABA transporter*$^+$neurons and found that vocalizations were never induced upon light stimulation, although breathing was altered in various ways (*Figure 5—figure supplement 1*). In summary, these data functionally demonstrate the existence of *Penk*$^+$*Slc17a6*$^+$ iRO neurons in adult mice and their ability to create vocalizations by modulating both breathing and presumably the larynx.

## Excitation of the iRO evoked nearly the entire murine lexicon

Above, we described that one mechanism for generating the different patterns of vocalizations was via the modulation of the breath airflow (positive intonation). Once again, this was defined as a positive correlation between expiratory airflow and pitch (*Figure 2*). We hypothesized that this property stems from the iRO's capacity to control breathing, and so we made the following predictions: (1) that the USVs evoked after stimulation would be biased to those with an endogenous positive correlation between airflow and pitch (like the down fm and step down), and (2) that the elicited USVs would be transformed to become more positively correlated.

We classified the evoked iRO vocalizations (*Penk*-Cre;*Slc17a6*-Flp;Cre$^{ON}$Flp$^{ON}$-ReaChR) with the CNN, and to our surprise, seven of the ten types of endogenous USVs were induced upon activation of the iRO (*Figure 5*). The most abundant elicited USV was the down fm which, in the endogenous dataset, had the strongest positive intonation (*Figure 5G* and *Figure 2*). Conversely, the USV with the strongest negative intonation was rarely found, up fm. These results are striking since the down fm is the least common endogenous USV and up fm is the most common (*Figure 1*). These results are consistent with the first prediction where the optically evoked USV types were biased toward those with endogenous positive intonation. Beyond this, all the ectopic USVs combined had a more positive association between airflow and pitch compared to all endogenous USVs (*Figure 5D*). This was not explained purely by the abundance of down fm since when these were omitted from the analysis the positive bias was unchanged. This aligns with the second prediction. These data demonstrate that the iRO is sufficient to pattern nearly all USV types, and that the pitch of the induced vocalizations tightly follows the breathing airflow.

## Discussion

Here, we propose that the intonation that establishes the diversity of the adult murine lexicon is explained by two mechanisms, the modulation of the breath waveform and presumably the size of the laryngeal opening. We describe that unique vocalization types have characteristic fluctuations in the expiratory airflow, whereby some changes in pitch are strongly correlated with airflow (positive intonation) while others are anti-correlated (negative intonation). These two mechanisms can even be used in the same breath to produce complex changes in pitch. To our surprise, seven of the ten USV types primarily used the positive intonation mechanism. These data support a novel and key role for the breathing system in the production of various types of vocalizations. In support of this hypothesis, we found re-activation of the breathing CPG during the expiration phase of the vocal breath. This ectopic activity appeared as a 'mini-breath' (inspiratory diaphragm then post-inspiratory laryngeal muscle activities) nested within a normal expiration and corresponded with modulated airflow and pitch. This resembles how the vocalization CPG, the iRO, patterns the rhythmic syllable structure of neonatal cry vocalizations. We show that the iRO is sufficient to induce most of the endogenous USV syllable types via the modulation of the breath airflow. In contrast to the natural lexicon, the pitch of the evoked USVs is primarily explained by positive intonation. These data imply that the iRO can produce the mechanism to pattern positive intonation, thereby suggesting that negative intonation derives from a separate neuronal component of the phonatory system. We propose that these two mechanisms can be used independently or in conjunction to generate the diverse repertoire of vocalizations (*Figure 5H*).

## The iRO likely patterns intonation for endogenous phonation

The description of the iRO within the adult neural circuit for phonation suggests a key role in patterning the endogenous adult vocalizations. In this case, we propose that the upstream PAG input would 'turn on' the iRO which then co-opts the breathing pacemaker and coordinates its anti-phase activity with the laryngeal muscles to produce and pattern the changes in breath airflow and vocal pitch (positive intonation). The iRO can do this since it is presynaptic to both the breathing pacemaker and the laryngeal motor neurons. In this case, the brief re-activation of inspiratory muscles we observed would slow ongoing expiration, enabling a decrease in airflow speed, and thus pitch. After, relaxation of these muscles results in an increase in expiration airflow and pitch. This type of oscillatory modulation we observed has also been seen in neonatal cries generated by the iRO (*Wei et al., 2022*). An important next step will be to validate the necessity of the iRO in adult phonation, as anticipated from its necessary role in neonatal vocalization. Nonetheless, the presence of the iRO across developmental stages implies a conserved role in innate vocalizations within the mouse and perhaps across the animal kingdom, where vocalization CPGs have been hypothesized and even identified in species from fish to birds to primates (*Zhang and Ghazanfar, 2020*; *Chagnaud et al., 2011*, *Hage, 2009b*, *Kelley et al., 2020*).

## The iRO can autonomously produce multiple vocalization patterns

A surprising finding is that ectopic activation of the iRO produces seven of the ten syllable types within the murine lexicon. How might this occur? One possibility is that the iRO has multiple modes which can each produce a different pattern of activity. Such a phenomenon has been demonstrated in other central pattern generating systems like the crustacean stomatogastric ganglia (*Marder and Bucher, 2001*; *Marder, 2012*). A more likely option is that additional mechanisms of vocal modulation are layered upon a basic pattern produced by the iRO. For example, other regions with direct control of the laryngeal motor neurons within RAm would add complexity to the vocalization induced by the iRO, akin to how vocal control by the human laryngeal motor cortex is conceptualized (*Figure 5H*; *Dichter et al., 2018*; *Silva et al., 2022*). Here, we propose that perhaps just two mechanisms (breath airflow and laryngeal opening) account for the intricacy of the murine sounds produced, and the layering of these enables a basic pitch structure within a breath to become sophisticated. Of note, these resulting models do not simplistically explain the origin of the pitch jumps that are present in some USV types and these may instead arise from mechanisms like an active process to alter the conformation of the larynx or upper airway.

Recently, glutamatergic laryngeal premotor neurons in RAm were identified that are sufficient to elicit vocalizations, albeit somewhat abnormal, and also necessary to produce sound (*Park et al., 2024*; *Veerakumar et al., 2023*). This raises the possibility that these RAm neurons compose the cellular basis for the laryngeal control mechanism we propose produces negative intonation (*Figure 5H*). However, the fact that ectopic excitation of these neurons or iRO elicits vocalizations suggests that reciprocal connections are engaged to create the dynamics of vocalizations. Consistently, iRO neurons project to RAm and vice versa (*Figure 3* and *Park et al., 2024*; *Veerakumar et al., 2023*). Interesting future studies will be to assess the necessity of either site to produce USVs upon ectopic excitation of the complementary region.

## The control of breathing airflow is a novel biomechanical mechanism for intonation

Intonation is a key aspect of communication, whereby the same word or phrase could be used as a question or a statement simply by different fluctuations in pitch. Additionally, in tonal languages, the same sound with differences in pitch can have completely different meanings. Our findings describe a novel biophysical mechanism for intonation and a cellular basis. Now, the iRO or direct modulation of breathing can serve as a starting point to map higher level components of brain-wide vocalization circuits that structure additional subliminal layers of perception in speech.

### Resource availability

#### Lead contact

Further information and requests for resources and reagents should be directed to and will be fulfilled by Kevin Yackle (kevin.yackle@ucsf.edu).

Materials availability

This study did not generate new unique reagents.

# Materials and methods

**Key resources table**

| Reagent type (species) or resource | Designation | Source or reference | Identifiers | Additional information |
|---|---|---|---|---|
| Genetic reagent (*Mus musculus*) | Slc17a6-IRES2-FlpO-D knock-in | The Jackson Laboratory | 030212 | |
| Genetic reagent (*M. musculus*) | Penk-IRES2-Cre | The Jackson Laboratory | 025112 | |
| Genetic reagent (*M. musculus*) | R26 LSL FSF ReaChR-mCitrine | The Jackson Laboratory | 024846 | |
| Genetic reagent (*M. musculus*) | Tac1-IRES2-Cre-D | The Jackson Laboratory | 021877 | |
| Genetic reagent (*M. musculus*) | Oprm1$^{Cre:GFP}$ KIKO | The Jackson Laboratory | 035574 | |
| Genetic reagent (*M. musculus*) | Vgat-ires-cre knock-in | The Jackson Laboratory | 028862 | |
| Genetic reagent (*M. musculus*) | Ai65(RCFL-tdT)-D | The Jackson Laboratory | 021875 | |
| Strain, strain background (adeno-associated virus) | AAV5-hSyn-Con/Fon-hChR2(H134R)-EYFP | Addgene, *Fenno et al., 2014*. | 55645-AAV5 | |
| Strain, strain background (adeno-associated virus) | AAV5-EF1a-DIO-hChR2(H134R)-EYFP-WPRE-HGHpA | Addgene | 20298-AAV5 | |
| Strain, strain background (adeno-associated virus) | AAVrg-EF1a-DIO-hChR2(H134R)-EYFP-WPRE-HGHpA | Addgene | 20298-AAVrg | |
| Antibody | Anti-GFP (chicken polyclonal) | Aves | GFP-1020 | (1:1000) |
| Antibody | Anti-ChAT (goat polyclonal) | Millipore | AB144p | (1:500) |
| Antibody | Anti-Chicken Alexa Fluor 488 (donkey polyclonal) | Invitrogen | A78948 | (1:500) |
| Antibody | Anti-Goat Alexa Fluor 546 (donkey polyclonal) | Invitrogen | A11056 | (1:500) |
| Antibody | Anti-Goat Alexa Fluor 647 (donkey polyclonal) | Invitrogen | A21447 | (1:500) |
| Software, algorithm | MATLAB | Mathworks | MATLAB 2022b | |
| Software, algorithm | VocalMat | *Fonseca et al., 2021*; *Fonseca and Santana, 2022* | https://github.com/ahof1704/VocalMat | |
| Software, algorithm | USVseg | *Tachibana et al., 2020*; *rtachi-lab, 2024* | https://github.com/rtachi-lab/usvseg | |

*Continued on next page*

| Reagent type (species) or resource | Designation | Source or reference | Identifiers | Additional information |
|---|---|---|---|---|
| | | *Continued* | | |
| Software, algorithm | Bespoke code | This paper, *Bachmutsky et al., 2020*, *Wei et al., 2022* | https://github.com/YackleLab/the-breath-shape-controls-intonation-of-mouse-vocalizations copy archived at *Yackle, 2024* | |
| Software, algorithm | Prism 9 | GraphPad | | |

## Experimental model and subject details

*Slc17a6*[FlpO] (*Daigle et al., 2018*), *Penk*[Cre] (*Tasic et al., 2018*), *Tac1*[Cre] (*Harris et al., 2014*), *Oprm1*[Cre] (*Liu et al., 2022*), *Vgat*[Cre] (*Vong et al., 2011*), Ai65 (*Madisen et al., 2015*), and LSL-FSF-ReaChR *Hooks et al., 2015* have been described. Mice were obtained from Jackson Laboratories and bred in-house at the University of California, San Francisco (UCSF) Laboratory Animal Research Center. Mice were housed in groups of two to five unless otherwise stated under a 12:12 light-dark cycle with ad libitum access to chow and water. All animal experiments were performed in accordance with national and Institutional Animal Care and Use Committee – University of California San Francisco guidelines with standard precautions to minimize animal stress and the number of animals used in each experiment. Protocol number AN195769.

### Recombinant viruses

All viral procedures followed the Biosafety Guidelines approved by the UCSF Institutional Animal Care and Use Program (IACUC) and Institutional Biosafety Committee (IBC). The viruses used in experiments were AAV5-hSyn-Con/Fon-hChR2(H134R)-EYFP (55645-AAV5, Addgene, $1.8×10^{13}$ vg/ml), AAV5-EF1a-DIO-hChR2(H134R)-EYFP-WPRE-HGHpA (20298-AAV5, Addgene, $1×10^{13}$ vg/ml), AAVrg-EF1a-DIO-hChR2(H134R)-EYFP-WPRE-HGHpA (20298-AAVrg, Addgene, $2.1×10^{13}$ vg/ml).

## Methods details

### Endogenous USV and breathing recording

Male *Slc17a6*[FlpO];*Penk*[Cre] mice (aged 8–16 weeks) were indi vidually housed and habituated to experimenter handling and a plethysmography chamber for >4 days. On the test day, the mice were placed in a clean cage base with a female mouse for 5 min and then moved to a plethysmography chamber. The chamber was modified to accommodate a microphone to record vocalizations (CM16/CMPA, Avisoft Bioacoustics) and the airflow in the chamber was measured by a spirometer (FE141, ADInstruments). Both data streams were acquired through a DAQ board (PCI-6251, National Instruments) and written to disk for offline analysis. Sound was acquired at 400 kHz and airflow at 1 kHz. After a 20 min habituation period, mice had airflow and sound recorded for 5 min before a cotton bud soaked in fresh urine was placed in the chamber, and sound and breathing were recorded for a further 15 min. Urine was collected the day of the experiment from a group of five female mice temporarily housed in a custom-made wire-bottom cage.

The recordings were run through VocalMat (*Fonseca et al., 2021*) for USV detection and only mice that produced >50 USVs in response to the stimulus were included for further analysis (6/14 mice). Airflow recordings were imported to MATLAB, high pass filtered (2 Hz), and smoothed. Breaths were taken from the first 200 s following urine presentation and features (Ti, Te, Pif, Pef, instantaneous frequency) were computed from segmented breaths as previously described (*Bachmutsky et al., 2020*). USV start and end times from VocalMat were used to identify which breaths contained USVs and calculate timing metrics (relative onset and offset from expiration onset and the same values normalized to expiratory duration). VocalMat was also used to identify the types of USV which were manually checked and corrected if necessary. For analysis of the relationship between airflow and frequency, a multitaper spectrogram was computed using code modified from USVseg (*Tachibana et al., 2020*) and then the frequency bin with the greatest power was taken from each time bin to create a vector of the peak frequency. The correlation coefficient of this peak frequency vector and the

expiratory airflow at the time stamps identified by VocalMat was then calculated for each identified USV.

For EMG recordings of the diaphragm and larynx we modified the protocol of *Hérent et al., 2020*. Electrodes were prepared from stainless steel (diaphragm, ground; AM Systems #793200) and tungsten (larynx; AM Systems #795500) wires and soldered to a 5-pin connector. An incision was made in the skin of the scalp and the fascia overlaying the skull was cleared. The twisted diaphragm electrodes were tunneled under the skin and inserted through the posterior ribs and sternum to cover the diaphragm as described in *Hérent et al., 2020*. To insert the larynx electrodes an incision was made in the neck and layers of muscle parted to expose the thyroid cartilage. Wires were then tunneled under the skin to the anterior incision, were inserted through the thyroid cartilage, and secured with superglue. A ground wire was inserted under the skin of the neck. The connector was then secured to the skull with dental cement, all incisions were closed with suture and mice were transferred to a heated recovery chamber.

After several days of recovery, mice were placed in the plethysmography chamber and the EMG pins were connected to an amplifier (AM Systems 1800). Audio and airflow signals were acquired as described above. EMG signals were bandpass filtered (300 Hz to 20 kHz) and acquired at 10 kHz. Diaphragm EMG recordings that included an ECG signal were manually annotated and the artifact removed from the trace. EMG traces were rectified and integrated offline with a modified Paynter filter in MATLAB. Three types of USVs (up fm, down fm, complex) were manually inspected and EMG peaks were annotated to generate histograms (*Figure 3*).

## Virus injection, fiber implantation, and optogenetics

Surgery was conducted with sterile tools and aseptic technique. Mice were first anesthetized with isoflurane (4%), the hair overlaying the scalp was shaved, and mice were placed in the stereotaxic frame where isoflurane (0.9–1.5%) was continuously delivered for the duration of the surgery. Mice were then injected with buprenorphine (0.1 mg/kg, s.c.) and carprofen (5 mg/kg, s.c.) and bupivicane (0.25 mg, under the skin of the scalp). The skin was then covered with betadine before an incision was made with a scalpel. The fascia was removed, and the skull dried with ethanol. The bregma and lambda sutures were identified and the skull was leveled using these landmarks. A craniotomy was drilled at the injection coordinates and a pulled glass pipette lowered to the injection site. An injection was made at a speed of 100 nl/min from an injection system (Nanoject III, Drummond). The injection pipette was left in place for 10 min following the injection then slowly retracted from the brain. In the case of bilateral injections, this process was then repeated on the other side. The skin was then closed by suture and the mouse transferred to a heated recovery cage.

For optogenetic experiments the virus injections were performed as described above. Once the injection pipette was removed the skull was scored with a scalpel blade and a fiber implant composed of a ferrule (CFLC230, Thorlabs) and an optic fiber (FT200EMT, Thorlabs) held in place with epoxy (F112, Thorlabs) inserted into the brain 200 µm dorsal to the injection site. The first fiber was glued in place while the second fiber was inserted. Once both fibers were in place, the skull was covered with dental cement (C&B Metabond) then a second layer of acrylic (Jet). After the skull cap was dried mice were transferred to a heated recovery cage. Coordinates (in mm) were as follows: iRO: 6.35 posterior to bregma, 5.4 ventral to skull surface, 1.2 lateral to midline; pBC: 6.73 posterior to bregma, 5.77 ventral to skull surface, 1.3 lateral to midline. To maximally excite the iRO system we did bilateral implants.

Mice were given 6 weeks between injection/implantation surgery and being used for experiments. ReaChR mice were implanted as described above. For optogenetic experiments bilateral fibers were connected to a split-patch cord (SBP(2)_200/220/900–0.37_m_FCM-2xZF1.25, Doric Lenses) and light was delivered from a laser (MBL-III-473, Opto Engine LLC) controlled by a TTL pulse generator (OTPG_4, Doric Lenses). Mice were placed in the plethysmography chamber with the microphone attached to simultaneously record breathing and sound along with the laser pulse commands. All three data streams were acquired through a DAQ board and written to disk for offline analysis. Sound was acquired at 250 kHz, airflow at 1 kHz, and laser pulse commands at 1 kHz. After a 20 min habituation period, laser pulses were delivered at frequencies of 5, 10, 20, and 50 Hz with pulse widths of 10, 25, or 50 ms for durations of 1 or 3 s. Laser power was adjusted to deliver ~20 mW of light at the patch cord tip although attenuation of light by the implanted fiber (determined post hoc) was variable

(12–21 mW). Each frequency/pulse width/duration combination was delivered five times with 7–9 s between presentations and a 30 s delay before the next stimulus was delivered.

Recordings were manually inspected for USVs during the laser epoch and recordings containing USVs were then run through VocalMat to find time stamps and to categorize each USV by type. MATLAB code was then used to quantify the correlation coefficients of optogenetically evoked USVs and the underlying airflow as described above. To analyze the breath statistics of optogenetically evoked breathing, the trial with stimulation parameters – 10 Hz, 25 ms pulse width, 3 s duration – was run through a code to extract breath statistics (Pif, Pef, instantaneous frequency) from the 30 s period prior to stimulation and from the five laser epochs.

## Histology

More than 6 weeks following viral injection or the completion of optogenetic testing mice were deeply anesthetized with isofluorane and transcardially perfused with 0.1 M phosphate-buffered saline (PBS) then PBS containing 4% paraformaldehyde (PFA). Brains were dissected from the fixed mice and refrigerated in 4% PFA overnight then cryoprotected in 30% sucrose in PBS. Brains were sectioned to 30 μm coronal on a freezing microtome. Sections were washed three times for 5 min in PBS before being incubated in blocking solution (PBS, 5% normal donkey serum, 0.3% Triton X-100) for 2 hr. Sections were then incubated overnight in primary antibodies (Chicken anti-GFP, 1:1000, Aves; Goat anti-ChAT, 1:500, Millipore) diluted in a carrier solution (PBS, 1% normal donkey serum, 0.3% Triton X-100). Following incubation sections were washed with PBS five times for 5 min then incubated in secondary antibodies (Donkey anti-Chicken 488, Donkey anti-Goat 546, Donkey anti-Goat 647) diluted 1:500 in a carrier solution (PBS, 0.3% Triton X-100) for 2 hr at room temperature. After secondary incubation, sections were washed with PBS five times for 5 min then mounted onto glass slides and cover-slipped with mounting media (Prolong Gold, Invitrogen) and 1 μg/ml DAPI.

## Quantification and statistical analysis

### Statistics

Data from MATLAB was imported to Prism 9 (GraphPad) for statistical analysis. For all statistical analysis except *Figure 2* and *Figure 5D–G* the mouse was used as the experimental unit. Data were assumed to be normally distributed and of equal variance and parametric tests were used. For data with one discrete variable and measurements made from the same animal (*Figure 1C–E*) paired t-test was used. For data with two variables one or both of which had more than two factors (*Figures 4G and 5C* and *Figure 1—figure supplement 2I and J*) two-way ANOVA was used with Sidak's post hoc test for multiple comparisons. To compare pitch-airflow correlations between observed and shuffled datasets (*Figure 2*, *Figure 5—figure supplement 1*), each USV was considered the experimental unit. For each USV type, we compared the observed values to two null distributions generated by shuffling one of the variables with one-way ANOVA and Sidak's post hoc test. To compare pitch-airflow correlations of endogenous and optically evoked USVs (*Figure 5D–G*) each USV was treated as the experimental unit since the vocal repertoire across animals was similar (*Figure 1F*) and simply taking a mean from each animal would under-represent the complexity of the data. For the comparison of correlation coefficients between optically evoked and endogenous USVs, two-way ANOVA with Sidak's post hoc test for two-way comparisons was used. p-Values below 0.05 were considered statistically significant.

## Acknowledgements

We thank Dr. YoonJeung Chang and Beatriz Cuevas for assistance with microscopy. We thank Dr. David Julius, and members of the Yackle lab for their input and revision of the manuscript. Funding: This work was supported by the Brain Initiative R34 NS127104, NINDS R01 NS126400, the Simon's Foundation, and the Klingenstein-Simons Award.

# Additional information

## Funding

| Funder | Grant reference number | Author |
|---|---|---|
| BRAIN Initiative | R34NS127104 | Kevin Yackle |
| National Institute of Neurological Disorders and Stroke | R01NS126400 | Kevin Yackle |
| Simons Foundation Autism Research Initiative | Pilot Award | Kevin Yackle |
| Esther A. and Joseph Klingenstein Fund | | Kevin Yackle |

The funders had no role in study design, data collection and interpretation, or the decision to submit the work for publication.

## Author contributions

Alastair MacDonald, Conceptualization, Software, Formal analysis, Investigation, Visualization, Methodology, Writing - original draft, Writing - review and editing; Alina Hebling, Xin Paul Wei, Investigation, Methodology, Writing - review and editing; Kevin Yackle, Conceptualization, Supervision, Funding acquisition, Writing - original draft, Project administration, Writing - review and editing

## Author ORCIDs

Alastair MacDonald (ID) http://orcid.org/0000-0002-2509-642X
Alina Hebling (ID) http://orcid.org/0009-0008-6247-9121
Xin Paul Wei (ID) http://orcid.org/0000-0002-6621-3143
Kevin Yackle (ID) http://orcid.org/0000-0003-1870-2759

## Ethics

All animal experiments were performed in accordance with national and Institutional Animal Care and Use Committee - University of California San Francisco guidelines with standard precautions to minimize animal stress and the number of animals used in each experiment. Protocol number AN195769.

Reviewer #1 (Public Review): https://doi.org/10.7554/eLife.93079.3.sa1
Reviewer #2 (Public Review): https://doi.org/10.7554/eLife.93079.3.sa2
Author response https://doi.org/10.7554/eLife.93079.3.sa3

# Additional files

## Supplementary files
• MDAR checklist

## Data availability

Source data files have been included as supplements to the corresponding figure. Data from all experiments has been deposited at Dryad (https://doi.org/10.5061/dryad.n8pk0p34d).

The following dataset was generated:

| Author(s) | Year | Dataset title | Dataset URL | Database and Identifier |
|---|---|---|---|---|
| MacDonald A, Hebling A, Wei XP, Yackle K | 2024 | Data from: The breath shape controls intonation of mouse vocalizations | https://doi.org/10.5061/dryad.n8pk0p34d | Dryad Digital Repository, 10.5061/dryad.n8pk0p34d |

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
