## [Editor Report · eLife assessment]

This **important** study examines the relationship between expiratory airflow and vocal pitch in adult mice during the production of ultrasonic vocalizations and also identifies a molecularly defined population of brainstem neurons that regulates mouse vocal production across development. The evidence supporting the study's conclusions that expiratory airflow shapes vocal pitch and that these brainstem neurons preferentially regulate expiratory airflow is novel and **compelling**. This work will be of interest to neuroscientists working on mechanisms and brainstem circuits that regulate vocal production and vocal-respiratory coordination.

---

## [Referee Report · Reviewer #1 (Public Review)]

Summary:

In this important work, the authors propose and test a model for the control of murine ultrasonic vocalizations (USV) in which two independent mechanisms involving changes in laryngeal opening or airflow control vocal tone. They present compelling experimental evidence for this dual control model by demonstrating the ability of freely behaving adult mice to generate vocalizations with various intonations by modulating both the breathing pattern and the laryngeal muscles. They also present novel evidence that these mechanisms are encoded in the brainstem vocalization central neural pattern generator, particularly in the component in the medulla called the intermediate reticular oscillator (iRO). The results presented clearly advance understanding of the developmental nature of the iRO, its ability to intrinsically generate and control many of the dynamic features of USV, including those related to intonation, and its coordination with/control of expiratory airflow patterns. This work will interest neuroscientists investigating the neural generation and control of vocalization, breathing, and more generally, neuromotor control mechanisms.

Strengths:

Important features and novelty of this work include:

(1) The study employs an effective combination of anatomical, molecular, and functional/ behavioral approaches to examine the hypothesis and provide novel data indicating that expiratory airflow variations can change adult murine USV's pitch patterns.

(2) The results significantly extend the authors' previous work that identified the iRO in neonatal mice by now presenting data that functionally demonstrates the existence of the critical Penk+Vglut2+ iRO neurons in adult mice, indicating that the iRO neurons maintain their function in generating vocalization throughout development.

(3) The results convincingly demonstrate that the iRO neurons encode and can generate vocalizations by modulating both breathing and the laryngeal muscles.

(4) The anatomical mapping and tracing results establish an important set of input and output circuit connections to the iRO, including input from the vocalization-promoting subregions of the midbrain periaqueductal gray (PAG), as well as output axonal projections to laryngeal motoneurons, and to the respiratory rhythm generator in the preBötzinger complex.

(5) These studies advance the important concept that the brainstem vocalization pattern generator integrates with the medullary respiratory pattern generator to control expiratory airflow, a key mechanism for producing various USV types characterized by different pitch patterns.

Weaknesses:

A limitation is that the cellular and circuit mechanisms by which the vocalization pattern generator integrates with the respiratory pattern generator to control expiratory airflow has not been fully worked out, requiring future studies.

---

## [Referee Report · Reviewer #2 (Public Review)]

Summary:

Both human and non-human animals modulate the frequency of their vocalizations to communicate important information about context and internal state. While regulation of the size of the laryngeal opening is a well-established mechanism to regulate vocal pitch, the contribution of expiratory airflow to vocal pitch is less clear. To consider this question, this study first characterizes the relationship between the dominant frequency contours of adult mouse ultrasonic vocalizations (USVs) and expiratory airflow using whole-body plethysmography. The authors also include data from a single mouse that combines EMG recordings from the diaphragm and larynx with plethysmography to provide evidence that the respiratory central pattern generator can be re-engaged to drive "mini-breaths" that occur during the expiratory phase of a vocal breath. Next, the authors build off of their previous work characterizing intermediate reticular oscillator (iRO) neurons in mouse pups to establish the existence of a genetically similar population of neurons in adults and show that artificial activation of iRO neurons elicits USV production in adults. Third, the authors examine the acoustic features of USV elicited by optogenetic activation of iRO and find that a majority of natural USV types (as defined by pitch contour) are elicited by iRO activation and that these artificially elicited USVs are more likely than natural USVs to be marked by positive intonation (positive relationship between USV dominant frequency and expiratory airflow).

Strengths:

Strengths of the study include the novel consideration of expiratory airflow as a mechanism to regulate vocal pitch and the use of intersectional methods to identify and activate the iRO in adult mice. The establishment of iRO neurons as a brainstem population that regulates vocal production across development is an important finding.

Weaknesses:

The conclusion that the respiratory CPG is re-engaged during "mini-breaths" throughout a given vocal breath would be strengthened by including analyses from more than one mouse.

---

## [Author Response]

The following is the authors’ response to the original reviews.

In the revised manuscript we have included an additional study that significantly contributes to the conclusions and models of the original version. Briefly, Figure 3 now describes our characterization of the diaphragm and laryngeal muscle activities (electromyography, EMG) during endogenous vocalizations. These EMGs also serve as representations of the brainstem breathing central pattern generator (CPG) inspiratory and post-inspiratory generating neurons, respectively. In our original submission, we found that many of the vocalizations had changes in pitch that mirrored the change in expiratory airflow (we termed positive intonation), and we proposed that the coordination of breathing muscles (like the inspiratory muscles) and larynx patterned this. This mechanism is akin to our findings for how neonatal cries are rhythmically timed and produced (Wei et al. 2022). The newly presented EMG data re-inforces this idea. We found that for vocalizations with positive intonation, the inspiratory diaphragm muscle has an ectopic burst(s) of activity during the expiration phase which corresponds to a decrease in airflow and pitch, and this is followed by laryngeal muscle activity and increased pitch. This can be cycled throughout the expiration to produce complex vocalizations with oscillations in pitch. A basal breath is hardwired for the laryngeal muscle activity to follow the diaphragm, so the re-cycling of this pattern nested within an expiration (a ‘mini-breath’ in a ‘breath’) demonstrates that the vocalization patterning system engages the entire breathing CPG. This contrasts with the canonical model that activity of the laryngeal premotor neurons control all aspects of producing / patterning vocalizations. Furthermore, this mechanism is exactly how the iRO produces and patterns neonatal vocalizations (Wei et al. 2022) and motivates the likely use of the iRO in adult vocalizations.

**Response to recommendations for the authors:**
**Reviewer #1**:(1) The authors should note in the Discussion that the cellular and circuit mechanisms by which the vocalization pattern generator integrates with the respiratory pattern generator to control expiratory airflow have not been fully worked out, requiring future studies.

This was noted in the discussion section “The iRO likely patterns intonation for endogenous phonation”.

(2) Please change the labeling of the last supplemental figure to Figure Supplemental 5.

Thank you for identifying this.

**Reviewer #2:**
Major concerns(1) While it is true that modulation of activity in RAm modulates the laryngeal opening, this statement is an incomplete summary of prior work. Previous studies (Hartmann et al., 2020; Zhang et al., 1992, 1995) found that activation of RAm elicits not just laryngeal adduction but also the production of vocal sounds, albeit vocal sounds that were spectrally dissimilar from speciestypical vocalizations. Moreover, a recent study/preprint that used an activity-dependent labeling approach in mice to optogenetically activate RAm neurons that were active during USV production found that re-activation of these neurons elicits USVs that are acoustically similar to natural USVs (Park et al., 2023). While the authors might not be required to cite that recent preprint (as it is not yet peer-reviewed), the fact that activation of RAm elicits vocal sounds is clear evidence that its effects go beyond modulating the size of the laryngeal opening, as this alone would not result in sound production (i.e., RAm activation must also recruit expiratory airflow). The authors should include these relevant studies in their Introduction. Moreover, the rationale for the model proposed by the authors (that RAm controls laryngeal opening whereas iRO controls expiratory airflow) is unclear with regard to these prior studies. The authors should include a discussion of how these prior findings are consistent with their model (as presented in the Introduction, as well as in Figure 4 and relevant Discussion) that RAm modulates the size of laryngeal opening but not expiratory airflow.

An introduction and discussion of the Veerakumar et. al. 2023 and Park et. al. 2024 manuscripts describing RAm in mice has now been included.

The iRO serves to coordinate the breath airflow and laryngeal adduction to produce sound and the intonation within it that mirrors the breath airflow. This occurs because the iRO can control the breathing CPG (synaptic input to the preBötC inspiratory pacemaker) and is premotor to multiple laryngeal muscles (Wei et. al. 2022). The modulation of the expiratory airflow is by inducing momentary contraction of the diaphragm (via excitation of the preBötC) which opposes (a.k.a. slows) expiration. This change in flow results in a decrease in pitch (Fig. 3 in the revised manuscript, Wei et. al. 2022).

It is our understanding that the basic model for RAm evoked USVs is that RAm evokes laryngeal adduction (and presumed abdominal expiratory muscle activation) and this activity is momentarily stopped during the breath inspiration by inhibition from the preBötC (Park et. al. 2024). So, in this basic model, any change in pitch and expiratory airflow would be controlled by tuning RAm activity (i.e., extent of laryngeal adduction). In this case, the iRO induced inspiratory muscle activity should not occur during expiration, which is not so (Fig. 3). Note, the activity of abdominal expiratory muscles during endogenous and RAm evoked USVs has not been characterized, so the contribution of active expiration remains uncertain. This is an important next step.

We have now included a discussion of this topic which emphasizes that iRO and RAm likely have reciprocal interactions (supported by the evidence of this anatomical structure). These interactions would explain why excitation of either group can evoke USVs and, perhaps, the extent that either group contributes to a USV explains how the pitch / airflow changes. An important future experiment will be to determine the sufficiency of each site in the absence of the other.

(2) The authors provide evidence that the relationship between expiratory airflow and USV pitch is variable (sometimes positive, sometimes negative, and sometimes not related). While the representative spectrograms clearly show examples of all three relationship types, no statistical analyses are included to evaluate whether the relationship between expiratory airflow and USV pitch is different than what one would expect by chance. For example, if USV pitch were actually unrelated to expiratory airflow, one might nonetheless expect spurious periods of positive and negative relationships. The lack of statistical analyses to explicitly compare the observed data to a null model makes it difficult to fully evaluate to what extent the evidence provided by the authors supports their claims.

We have now included two null distributions and compared our observed correlation values to these. The two distributions were created by taking each USV / airflow pair and randomly shuffling either the normalized USV pitch values (pitch shuffled) or the normalized airflow values (airflow shuffled) to simulate the distribution of data should no relationship exist between the USV pitch and airflow.

(3) The relationship between expiratory airflow and USV pitch comes with two important caveats that should be described in the manuscript. First, even in USV types with an overall positive relationship between expiratory airflow and pitch contour, the relationship appears to be relative rather than absolute. For example, in Fig. 2E, both the second and third portions of the illustrated two-step USV have a positive relationship (pitch goes down as expiratory airflow goes down). Nonetheless, the absolute pitch of the third portion of that USV is higher than the second portion, and yet the absolute expiratory airflow is lower. The authors should include an analysis or description of whether the relationship between expiratory airflow and USV pitch is relative vs.

absolute during periods of 'positive intonation'.

The relationship between pitch and airflow is relative and this in now clarified in the text. To determine this, we visualized the relationship between the two variables by scatterplot for each of the USVs syllables and, as the reviewer notes, a given airflow cannot predict the resulting frequency and vice versa.

(4) A second important caveat of the relationship between expiratory airflow and USV pitch is that changes in expiratory airflow do not appear to account for the pitch jumps that characterize mouse USVs (this lack of relationship also seems clear from the example shown in Fig. 2E). This caveat should also be stated explicitly.

The pitch jumps do not have a corresponding fluctuation in airflow, and this is now stated in the results and discussion.

(5) The authors report that the mode of relationship between expiratory airflow and USV pitch (positive intonation, negative intonation, or no relationship) can change within a single USV. Have the authors considered/analyzed whether the timing of such changes in the mode of relationship coincides with pitch jumps? Perhaps this isn’t the case, but consideration of the question would be a valuable addition to the manuscript.

We analyzed a subset of USVs with pitch jumps that were defined by a change >10 kHz, at least 5ms long, and had one or two jumps. The intonation relationships between the sub-syllables within a USV type were not stereotyped as evidenced by the same syllable being composed of combinations of both modes.

(6) The authors incorrectly state that PAG neurons important for USV production have been localized to the ventrolateral PAG. Tschida et al., 2019 report that PAG-USV neurons are located predominantly in the lateral PAG and to a lesser extent in the ventrolateral PAG (see Fig. 5A from that paper). The finding that iRO neurons receive input from VGlut2+ ventrolateral PAG neurons represents somewhat weak evidence that these neurons reside downstream of PAG-USV neurons. This claim would be strengthened by the inclusion of FOS staining (following USV production), to assess whether the Vglut+ ventrolateral PAG neurons that provide input to iRO are active in association with USV production.

This comment correctly critiques that our PAG à iRO tracing does not demonstrate that the labeled PAG neurons are sufficient nor necessary for vocalization. Directly demonstrating that activation and inhibition the PAG-iRO labeled neurons ectopically drives or prevents endogenous USVs is an important next step. While FOS implies this connectivity, it does not definitely establish it and so this experiment is impacted by some of the caveats of our tracing (e.g. PAG neurons that drive sniffing might be erroneously attributed to vocalization).

Our reading of the literature could not identify an exact anatomical location within the mouse PAG and this site appears to vary within a study and between independent studies (like within and between Tschida et. al. 2019 and Chen et. al. 2021). The labeling we observed aligns with some examples provided in these manuscripts and with the data reported for the retrograde tracing from RAm (Tschida et al 2019).

(7) In Figure S5A, the authors show that USVs are elicited by optogenetic activation of iRO neurons during periods of expiration. In that spectrogram, it also appears that vocalizations were elicited during inspiration. Are these the broadband vocalizations that the authors refer to in the Results? Regardless, if optogenetic activation of iRO neurons in some cases elicits vocalization both during inspiration and during expiration, this should be described and analyzed in the manuscript.

The sound observed on the spectrogram during inspiration is an artefact of laser evoked head movements that resulted in the fiber cable colliding with the plethysmography chamber. In fact, tapping an empty chamber yields the same broad band spectrogram signal. The evoked USV or harmonic band vocalization is distinct from this artefact and highlighted in pink.

(8) Related to the comment above, the authors mention briefly that iRO activation can elicit broadband vocalizations, but no details are provided. The authors should provide a more detailed account of this finding.

The broadband harmonic vocalizations we sometimes observe upon optogenetic stimulation of AAV-ChR2 expressing iRO neurons are akin to those previously described within the mouse vocal repertoire (see Grimsley et. al .2011). We have added this citation and mentioned this within the text.

(9) The effects of iRO stimulation differ in a couple of interesting ways from the effects of PAGUSV activation. Optogenetic activation of PAG-USV neurons was not found to entrain respiration or to alter the ongoing respiratory rate and instead resulted in the elicitation of USVs at times when laser stimulation overlapped with expiration. In contrast, iRO stimulation increases and entrains respiratory rate, increases expiratory and inspiratory airflow, and elicits USV production (and also potentially vocalization during inspiration, as queried in the comment above). It would be informative for the authors to add some discussion/interpretation of these differences.

We have added a section of discussion to describe the how these different results may be explained by the iRO being a vocal pattern generator versus the PAG as a ‘gating’ signal to turn on the medullary vocalization patterning system (iRO and RAm). See discussion section ‘The iRO likely patterns intonation for endogenous phonation’.

(10) The analysis shown in Fig. 4D is not sufficient to support the author’s conclusion that all USV types elicited by iRO activation are biased to have more positive relationships between pitch and expiratory airflow. The increase in the relative abundance of down fm USVs in the opto condition could account for the average increase in positive relationship when this relationship is considered across all USV types in a pooled fashion. The authors should consider whether each USV type exhibits a positive bias. Although such a comparison is shown visually in Fig. 4G, no statistics are provided. All 7 USV types elicited by optogenetic activation of iRO should be considered collectively in this analysis (rather than only the 5 types currently plotted in Fig. 4G).

In the original submission the statistical analysis of r values between opto and endogenous conditions was included in the figure legend (‘panels E-G, two-way ANOVA with Sidak’s post-hoc test for two-way comparisons was used; all p-values > 0.05), and this has not changed in the revised manuscript. We have now provided the suggested comparison of opto vs endogenous USVs without down fm (Fig. 5D). This positive shift in r is statistically significant (…).

(11) The evidence that supports the author’s model that iRO preferentially regulates airflow and that RAm preferentially regulates laryngeal adduction is unclear. The current study finds that activation of iRO increases expiratory (and inspiratory) airflow and also elicits USVs, which means that iRO activation must also recruit laryngeal adduction to some extent. As the authors hypothesize, this could be achieved by recruitment of RAm through iRO’s axonal projections to that region.

Note, it is more likely that iRO is directly recruiting laryngeal adduction as they are premotor to multiple laryngeal muscles like the thyroarytenoid and cricothyroid (Wei et. al. 2022). The ‘Discussion’ now includes our ideas for how the iRO and RAm likely interact to produce vocalizations.

In the recent preprint from Fan Wang’s group (Park et al., 2023), those authors report that RAm is required for USV production in adults, and that activation of RAm elicits USVs that appear species-typical in their acoustic features and elicits laryngeal adduction (assessed directly via camera). Because RAm activation elicits USVs, though, it must by definition also recruits expiratory airflow. Can the authors add additional clarification of how the evidence at hand supports this distinction in function for iRO vs RAm?

See response to ‘Major Concern #1”.

Minor concerns(1) The authors might consider modifying the manuscript title. At present, it primarily reflects the experiments in Figure 2.

We have provided a title that we feel best reflects the major point of the manuscript. We hope that this simplicity enables it to be recognized by a broad audience of neuroscientists as well as specialists in vocalization and language.

(2) The statement in the abstract that "patterns of pitch are used to create distinct 'words' is somewhat unclear. Distinct words are by and large defined by combinations of distinct phonemes. Are the authors referring to the use of "tonemes" in tonal languages? If so, a bit more explanation could be added to clarify this idea. This minor concern includes both the Abstract, as well as the first paragraph of the Introduction.

We have clarified this line in the abstract to avoid the confusing comparison between mouse vocalizations and human speech. In the introduction we have expanded our explanation to clarify that variations in pitch are a component of spoken language that add additional meaning and depth to the underlying, phonemic structure.

(3) Multiple terms are used throughout the manuscript to refer to expiratory airflow: breath shape (in the title), breath pattern, deviations in exhalation, power of exhalation, exhalation strength, etc. Some of these terms are vague in meaning, and a consolidation of the language would improve the readability of the abstract and introduction.

We have chosen a smaller selection of descriptive words to use when describing these breath features.

(4) Similarly, "exhalation" and "expiration" are both used, and a consistent use of one term would help readability.

See point 3.

(5) In a couple of places in the manuscript, the authors seem to state that RAm contains both laryngeal premotor neurons as well as laryngeal motor neurons. This is not correct to our knowledge., but if we are mistaken, we would ask that the authors add the relevant references that report this finding.

It is our understanding that the RAm is defined as the anatomical region consistent with the murine rostral and caudal ventral respiratory groups composed of multiple premotor neuron pools to inspiratory, expiratory, laryngeal, and other orofacial muscles. This is supported by neurons within RAm that reflect multiple phases of the inspiratory and expiratory cycle (Subramanian et. al. 2018) and excitation of sub-regions within RAm modulating multiple parts of the breathing control system (Subramanian et. al. 2018 and Subramanian 2009). Rabies tracing of the various premotor neurons which define the anatomical region of RAm in the mouse shows that they surround the motor neurons in the loose region of the nucleus ambiguus (the anatomical location of RAm) for multiple muscles of the upper airway system, such as the thyroarytenoid (Wu et. al. 2017, Dempsey et. al. 2021 and Wei et. al. 2022). Given that the name RAm reflects a broad anatomical location, we have used it to describe both the premotor and motor neurons embedded within it. We have now clarified this in the text.

(6) The statistical analysis applied in Figure 1C is somewhat confusing. The authors show two distributions that appear different but report a p-value of 0.98. Was the analysis performed on the mean value of the distributions for each animal, the median, etc.? If each animal has two values (one for USV+ breaths and one for USV- breaths), why not instead compare those with a paired t-test (or Wilcoxon rank sign)? Additional information is needed to understand how this analysis was performed.

The original manuscript version used a two-way anova to compare the normalized histogram of instantaneous frequency for breaths with (USV+) or without (USV-) for each animal (first factor: USV+/-, second factor: Frequency). The p-value for the first factor (USV) was 0.98 showing no statistically significant effect of USV on the distribution of the histogram.

For simplicity, we have instead performed the analysis as suggested and include a bar graph. This analysis shows that the instantaneous frequency of USV breaths is, in fact, statistically significantly lower than those without USVs. We have updated the figure legend and text to reflect this.

(7) The use of the word "syllable" to describe parts of a USV that are produced on a single breath may be confusing to some scientists working on rodent USVs. The term 'syllable' is typically used to describe the entirety of a USV, and the authors appear to use the term to describe parts of a USV that are separated by pitch jumps. The authors might consider calling these parts of USVs "sub-syllables".

We have clarified these descriptions throughout the text. We now refer to the categories as ‘syllable types’, define ‘syllables’ as ‘a continuous USV event’ with no more than 20ms of silence within and finally ‘sub-syllables’ to refer to components of the syllable separated by jumps in frequency (but not gaps in time).

(8) In Figure S3, final row, the authors show a USV produced on a single breath that contains two components separated by a silent period. This type of bi-syllabic USV may be rare in adults and is similar to what the authors showed in their previous work in pups (multiple USVs produced on a single expiration, separated by mini-inspirations). One might assume that the appearance of such USVs in pups and their later reduction in frequency represents a maturation of vocalrespiratory coordination. Nonetheless, the appearance of bi-syllabic USVs has not been reported in adult mice to our knowledge, and the authors might consider further highlighting this finding.

We were also struck by the similarity of these USVs to our study in neonates and such types of similarities sparked an interest in the role of the iRO in patterning adult USVs. We now include a description of the presence and abundance of bi- and tri-syllablic calls observed in our recordings to highlight this finding.

(9) Figure 4 is referenced at the end of the second Results section, but it would seem that the authors intended to reference Figure 2.

For simplicity we included some of the referenced data within Fig. S5. We appreciate the recommendation.

(10) In the optogenetic stimulation experiments, the authors should clarify why bilateral stimulation was applied. Was unilateral stimulation ineffective or less effective? The rationale provided for the use of bilateral stimulation (to further localize neural activation) is unclear.

The iRO is bilateral and, we presume, functions similarly. So, we attempted to maximally stimulate the system. We have clarified this in the methods.

(11) Figure Supplemental '6' should be '5'.

Thanks!

(12) Last sentence of the Introduction: "Lasty" should be "lastly".

Thanks!

(13) There are two references for Hage et al., 2009. These should be distinguished as 2009a and 2009b for clarity.

Thanks!